# Multi-Level Genetic Variation and Selection Strategy of *Neolamarckia cadamba* in Successive Years

Qingmin Que [1,2,3], Chunmei Li [1,2,3], Buye Li [1,2,3], Huiyun Song [1,2,3], Pei Li [1,2,3], Ruiqi Pian [1,2,3], Huaqing Li [4], Xiaoyang Chen [1,3,*] and Kunxi Ouyang [2,*]

1 State Key Laboratory for Conservaion and Utilization of Subtropical Agro-Bioresources, South China Agricultural University, Wushan Road 483, Tianhe District, Guangzhou 510642, China; qmque@scau.edu.cn (Q.Q.); dachunli134@gmail.com (C.L.); byl766798@gmail.com (B.L.); hysong@stu.scau.cn (H.S.); lipei-meinv@163.com (P.L.); rqpian2003@126.com (R.P.)
2 Guangdong Key Laboratory for Innovative Development and Utilization of Forest Plant Germplasm, South China Agricultural University, Wushan Road 483, Tianhe District, Guangzhou 510642, China
3 Guangdong Province Research Center of Woody Forage Engineering Technology, College of Forestry and Landscape Architecture, South China Agricultural University, Wushan Road 483, Tianhe District, Guangzhou 510642, China
4 China Forestry Group Leizhou Forestry Bereau Co., Ltd., Zhanjiang 524348, China; huaqiangli3@gmail.com
* Correspondence: xychen@scau.edu.cn (X.C.); kxouyang@scau.edu.cn (K.O.)

**Abstract:** *Neolamarckia cadamba* (Roxb.) Bosser is a tropical evergreen broadleaf tree species that could play an important role in meeting the increasing demand for wood products. However, multi-level genetic variation and selection efficiency for growth traits in *N. cadamba* is poorly characterized. We therefore investigated the efficiency of early selection in *N. cadamba* by monitoring the height (HT), diameter at breast height (DBH), and tree volume (V) in 39 half-sib families from 11 provenances at ages 2, 3, 4, 5, and 6 years in a progeny test. Age-related trends in growth rate, genetic parameters in multi-level, efficiency of early selection, and realized gain in multi-level for growth traits were analyzed. The result showed that genetic variation among families within provenances was higher than that among provenances. The estimated individual heritability values for the growth traits ranged from 0.05 to 0.26, indicating that the variation of growth traits in *N. cadamba* was subject to weak or intermediate genetic control. The age–age genetic correlations for growth traits were always positive and high (0.51–0.99), and the relationships between the genetic/phenotypic correlations and the logarithm of the age ratio (*LAR*) were described well by linear models ($R^2 > 0.85$, except the fitting coefficient of genetic correlation and *LAR* for HT was 0.35). On the basis of an early selection efficiency analysis, we found that it is the best time to perform early selection for *N. cadamba* at age 5 before half-rotation, and the selection efficiencies were 157.28%, 151.56%, and 127.08% for V, DBH, and HT, respectively. Higher realized gain can be obtained by selecting superior trees from superior families. These results can be expected to provide theoretical guidance and materials for breeding programs in *N. cadamba* and can even be a reference for breeding strategies of other fast-growing tree species.

**Keywords:** genetic variability; growth trait; *Neolamarckia cadamba*; early selection



## 1. Introduction

The growth of the world's population has been accompanied by contractions of the areas of native and/or natural forests protected by governments, and therefore tree plantations and agroforestry have become vital for meeting the increasing demand for pulp, paper, and wood products [1]. The economic and commercial values of tree plantations are directly affected by the quantity and quality of wood produced [2], which are the most important objectives for forest tree breeding. The harvest efficiency can be low for trees due to their long period of rotation and usually varying genetic backgrounds [3]. It is therefore desirable to shorten the period of rotation. Early or indirect selection of trees

is one important way of shortening the timber production cycle, increasing the potential efficiency of improvement with genetic approaches, and maximizing annual production [3].

In order for the efficiency of early selection to be increased, it is necessary to determine the genetic variation of trees at different levels and the optimum age of early selection. Different levels of genetic variation determine the breeding strategy of a forest tree species [4]. The various growth traits of different tree species are subject to different levels of genetic control. For example, heritability for biomass of pedunculate oak (*Quercus robur* L.) were the highest for provenances, average for families, and the lowest for individual trees (0.18) [4]. In contrast, the genetic variation of tree volume for eucalyptus (*Eucalyptus saligna* Sm.) between families was generally greater than that between provenances [5]. Furthermore, the determination of the efficiency for early selection requires an analysis of age trends in genetic parameters, such as heritability and age–age genetic/phenotypic correlations [6]. Age–age correlations and the effectiveness of early selection have been extensively reported for a large number of tree species. Wu (2019) found that selecting Chinese fir (*Cunninghamia lanceolata* (Lamb.) Hook.) trees aged between 6 and 12 years can improve the genetic gain for growth traits. It was efficient for early selection of growth traits at one-third of the rotation period in sugi (*Cryptomeria japonica* (L. f.) D. Don) [7], as well as for that of microfibril angle at the age of 4 years and modulus of elasticity between 5 and 8 years for density in lodgepole pine (*Pinus contorta* Dougl. ex Loud. var. *latifolia* Engelm.) [8].

*Neolamarckia cadamba* (Roxb.) Bosser (Rubiaceae) is a tropical evergreen broadleaf tree species belonging to the Rubiaceae family that is widely distributed in South and South East Asia. It has been cultivated and introduced to Puerto Rico, Surinam, Venezuela, South Africa, Costa Rica, and other tropical and subtropical countries [9] due to its great economic and ecological value [10]: its timber is used for furniture production and light construction, as well as pulp, paper production, plywood, and veneer [11]. The leaf, bark, flowers, and fruits of *N. cadamba* are also used widely in traditional Indian ethno-medicine and modern medical practice [12]. Additionally, its fruits are used for juice [13], its pollen is a food source for honey bees [14], its leaves are converted to silage [15], and the whole tree is used in landscaping [16]. Because of *N. cadamba* commercial importance, breeders search to improve its growth traits and wood properties [11,17–19]. Importantly, as a fast-growing tree species, *N. cadamba* can attain a height of almost 18 m and a DBH of 25 cm at the age of 9 years under normal conditions [20]. Thus, it has been described as "a miraculous tree" [21] and may be a good alternative tree species for cultivation in suitable regions to meet the increasing demand for wood products. Therefore, it is important to study the genetic variation of *N. cadamba* for further selection and breeding in the future. Although many provenance tests have been conducted on *N. cadamba* [9,20], only a few have addressed the multi-level genetic variation or early selection efficiency.

Progeny test on *N. cadamba* from China were established in Guangdong province in 2014. Early results from these field tests revealed significant differences in height and DBH among provenances or families within provenances, with heritability values between 0.53 and 0.79 [22,23]. However, the variation of selection efficiency with age and intrafamilial correlations in the field test have not been reported previously. This work therefore aims to (i) explore the weight and dynamic changes of genetic variation structure at different levels in *N. cadamba*, (ii) identify age trends in heritabilities and age–age correlations for *N. cadamba*, (iii) determine the efficiency of early selection in *N. cadamba*, and (iv) explore improvement strategy of *N. cadamba* with multiple levels of variation. We found that genetic variation among families within provenances was higher than that among provenances and determined the optimal selection age before half-rotation The results will be used to develop appropriate selection strategies for *N. cadamba* breeding programs in southern China, providing a reference for the breeding strategies of other fast-growing tree species.

## 2. Materials and Methods

### 2.1. Materials

Data were collected from a six-year-old *N. cadamba* progeny trial established in Leizhou, Guangdong province (21°10′06″ N, 110°21′34″ E), with mean annual temperature of 22 °C and annual rainfall of 1711.6 mm. The maximum and minimum monthly average temperatures at the trial site during the studied period were 28.4 °C and 15.5 °C, respectively. The trail was established in the spring of 2014. The field design consisted of complete randomized blocks with 10 replications and 5 trees per subdivided plots in a 3 × 3 m square spacing. Thirty-nine families were planted in the experimental trial. Of these, 39 families were from open-pollinated seeds of 10 natural populations of China distributed in Yunnan, Guangxi, and Guangdong, and 1 natural population from Indonesia (Table 1). The sampled trees were selected on the basis of being phenotypically average or above in terms of stem diameter at breast height and total height, compared with neighboring trees in the population. The distance among mother trees within the population was kept at least at 100 m to minimize genetic relatedness between seed lots. Seeds were collected by climbing the selected trees. Each seed lot was separated for individual sample labelled with the location information.

**Table 1.** Geographical locations of the sampled *N. cadamba* populations and their climatic properties.

| Provenance | Family Number | Latitude(°N) | Longitude (°E) | Altitude (m) | Average Annual Temperature (°C) | Minimum Temperature (°C) | Maximum Temperature (°C) | Frostless Period (d) | Average Annual Precipitation (mm) |
|---|---|---|---|---|---|---|---|---|---|
| GXLZ | 1 | 22.36 | 106.84 | 269 | 22.2 | 0.8 | 39.9 | 352 | 1260 |
| GXFCG | 1 | 21.77 | 107.35 | 235 | 21.8 | 1.4 | 37.8 | 360 | 2512 |
| GXNN | 2 | 22.85 | 108.40 | 80 | 21.7 | −2.4 | 40.4 | 364 | 1304 |
| GDGZ | 3 | 23.10 | 113.21 | 10 | 22.1 | 0.0 | 39.3 | 346 | 1696 |
| GDYF | 1 | 22.10 | 112.02 | 346 | 21.5 | −1.0 | 39.1 | 345 | 1670 |
| YNBS | 6 | 25.08 | 99.16 | 1670 | 17.4 | −4.2 | 40.4 | 283 | 1710 |
| YNDH | 7 | 24.08 | 97.39 | 780 | 18.9 | −2.9 | 35.7 | 299 | 1544 |
| YNJH | 11 | 21.02 | 101.04 | 552 | 21.0 | 2.7 | 41.1 | 365 | 1197 |
| YNMS | 2 | 24.20 | 98.95 | 913 | 19.6 | −0.6 | 36.2 | 315 | 1650 |
| YNMN | 4 | 21.40 | 101.30 | 631 | 21.0 | 0.5 | 38.4 | 331 | 1540 |
| IDN | 1 | −0.81 | 102.38 | - | 25.0 | - | - | 365 | 2546 |

### 2.2. Data Collection

Diameter at breast height (DBH in cm, 1.3 m above ground level) and height (HT in m) were measured for all individuals from 2 to 6 years after planting. The volume of each individual tree (V in m$^3$) was calculated according to the following formula [20]:

$$V = 3.69 \times 10^{-5} \times DBH^2 \times HT \tag{1}$$

The traits analyzed in this study are denoted by DBH-2, HT-3, V-4, etc., with the number in each case indicating the age at which the corresponding trait was measured.

### 2.3. Statistical Analysis

Variance and covariance components for genetic analyses were estimated using the SAS Mixed procedure and PROC VARCOMP (method REML) in SAS software version 9.1.3 (SAS Institute Inc., Cary, NC, USA), on the basis of a mixed linear model:

$$y_{ijk} = \mu + B_i + P_j + PB_{ij} + F(P)_{jk} + F(P)B_{ijk} + e_{ijkl} \tag{2}$$

where $y_{ijk}$ is the observations of traits; $\mu$ is the overall mean; $B_i$ is the fixed effects of block; $P_j$ and $F(P)_{jk}$ are the random effects of provenance and family within provenance, respectively; $PB_{ij}$ and $F(P)B_{ijk}$ are the random effects of provenance–block and family–block, respectively; and $e_{ijkl}$ is residual error (the random effect of individual tree within plot).

Individual heritability ($h^2$) was calculated according to the following equations on the basis of the variance component estimates from the model analyses:

$$h^2 = \frac{4V_F}{V_F + V_{FB} + V_e} \tag{3}$$

Here, $V_F$, $V_{FB}$, and $V_e$ represent the family within provenance, family–block interaction, and random error variance, respectively.

The coefficient of genetic variation ($CV_G$) was calculated with the following formulas [24,25]:

$$CV_G(\%) = 100 \times \frac{\sqrt{V_F + V_P}}{\overline{X}} \tag{4}$$

where $\overline{X}$, $V_F$, and $V_P$ represent the mean value of each growth trait, family variance and provenance variance, respectively.

The additive genetic correlation and phenotypic correlation between the growth traits at different ages were calculated as

$$r_A = \frac{Cov_{a_1,a_2}}{\sqrt{V_{a_1} \times V_{a_2}}} \tag{5}$$

$$r_P = \frac{Cov_{a_1,a_2} + Cov_{e_1,e_2}}{\sqrt{\left(V_{a_1} + V_{e1}\right) \times \left(V_{a_2} + V_{e2}\right)}} \tag{6}$$

Here, $r_A$ and $r_P$ are the genetic and phenotypic correlations, respectively; $Cov_{a_1,a_2}$ is the additive genetic covariance between the trait in question at ages $a_1$ and $a_2$ ($a_1 < a_2$); $Cov_{e_1,e_2}$ is the error covariance between the same trait at ages $a_1$ and $a_2$; $V_{a_1}$ and $V_{a_2}$ are the additive genetic variances at ages $a_1$ and $a_2$, respectively; and $V_{e_1}$ and $V_{e_2}$ are the residual error variances at ages $a_1$ and $a_2$, respectively.

Natural logarithms of age ratios ($LAR$) [26] were calculated using the following formula:

$$LAR = log_e(age_{young} - age_{old}) \tag{7}$$

where $age_{young}$ and $age_{old}$ represent the younger and older ages of the targeted age pair, respectively. To determine how age affected genetic correlations, we performed linear regression analyses between $r_A/r_P$ and $LAR$.

When evaluating early selection efficiency, we used the growth traits at age 6 (HT-6, DBH-6, and V-6) as the target traits. Under the assumption that the intensity of selection at the rotation age was equal to that at younger ages, we computed the selection efficiency ($SE_{GPY}$) as [27]

$$SE_{GPY} = \frac{r_{A_{a_1,a_2}} h_{a_1} (a_2 + t)}{h_{a_2} (a_1 + t)} \tag{8}$$

where $a_1$ and $a_2$ are the ages corresponding to the juvenile and late trait measurements ($a_2$ was always 6 in this work), respectively; $t$ is the additional number of years required to complete the rotation cycle [28], which was 6 years in this work; $r_A$ is the additive genetic correlation between the two ages; and $h_{a1}$ and $h_{a2}$ are the square roots of the heritability at the juvenile and late ages, respectively.

Genetic gains ($\Delta G$) and realized gains ($\Delta Gr$) were calculated as follows [29]:

$$\Delta G = \frac{ih^2\sigma}{\overline{X_S}} \times 100\% \tag{9}$$

$$\Delta Gr = \frac{\overline{x}}{\overline{X}} \times 100\% \tag{10}$$

where $i$ is selection intensity (in the present study, the selection intensity of individual selection is three times of the standard deviation in each trait of the superior family); $h^2$ is heritability of trait; $\sigma$ is standard deviation of the superior family; $\overline{x}$ is mean of traits for selected provenances or families; and $\overline{X}$ is mean of traits for all provenances or families. $\overline{X_S}$ is mean of traits for all selected families.

## 3. Results

### 3.1. Phenotypic Variation

As Table 2 shows, between the ages of 2 and 6 years, the individual mean values of HT increased by 8.34 m (from 3.86 to 12.20 m), plus 8.89 cm (from 2.75 to 11.64 cm) and 0.06651 m$^3$ (from 0.00179 to 0.0683 m$^3$) for those of DBH and V increase, respectively. In addition, the mean annual increments in HT, DBH, and V were 2.09 m, 2.22 cm, and 0.0.0166 m$^3$, respectively.

**Table 2.** Descriptive statistics for growth traits at different ages in the progeny test population of *N. cadamba.*

| Trait | Mean | Minimum | Maximum | Range | Standard Error (SE) | Coefficient of Variation (CV)/% |
|-------|------|---------|---------|-------|---------------------|----------------------------------|
| HT2 (m) | 3.86 | 2.00 | 7.40 | 5.40 | $2.71 \times 10^{-2}$ | 27.80 |
| HT3 (m) | 7.74 | 2.90 | 12.20 | 9.30 | $4.52 \times 10^{-2}$ | 23.98 |
| HT4 (m) | 9.74 | 3.50 | 14.90 | 11.40 | $5.24 \times 10^{-2}$ | 21.99 |
| HT5 (m) | 10.86 | 5.10 | 16.60 | 11.50 | $5.84 \times 10^{-2}$ | 21.20 |
| HT6 (m) | 12.20 | 6.80 | 18.20 | 11.40 | $6.45 \times 10^{-2}$ | 20.49 |
| DBH2 (cm) | 2.75 | 0.80 | 5.20 | 4.40 | $1.62 \times 10^{-2}$ | 23.37 |
| DBH3 (cm) | 5.45 | 1.70 | 8.50 | 6.80 | $2.80 \times 10^{-2}$ | 21.11 |
| DBH4 (cm) | 7.78 | 4.10 | 10.80 | 6.70 | $3.07 \times 10^{-2}$ | 16.12 |
| DBH5 (cm) | 10.36 | 6.50 | 13.70 | 7.20 | $3.87 \times 10^{-2}$ | 14.72 |
| DBH6 (cm) | 11.64 | 8.50 | 15.80 | 7.30 | $3.51 \times 10^{-2}$ | 11.70 |
| V2 (m$^3$) | $1.79 \times 10^{-3}$ | $1.79 \times 10^{-4}$ | $1.03 \times 10^{-2}$ | $1.01 \times 10^{-2}$ | $3.38 \times 10^{-5}$ | 74.46 |
| V3 (m$^3$) | $1.37 \times 10^{-2}$ | $5.65 \times 10^{-4}$ | $4.25 \times 10^{-2}$ | $4.19 \times 10^{-2}$ | $1.91 \times 10^{-4}$ | 57.58 |
| V4 (m$^3$) | $2.98 \times 10^{-2}$ | $2.00 \times 10^{-3}$ | $7.84 \times 10^{-2}$ | $7.64 \times 10^{-2}$ | $3.73 \times 10^{-4}$ | 51.22 |
| V5 (m$^3$) | $4.89 \times 10^{-2}$ | $6.49 \times 10^{-3}$ | $1.29 \times 10^{-1}$ | $1.23 \times 10^{-1}$ | $6.19 \times 10^{-4}$ | 49.90 |
| V6 (m$^3$) | $6.83 \times 10^{-2}$ | $1.45 \times 10^{-2}$ | $1.51 \times 10^{-1}$ | $1.37 \times 10^{-1}$ | $8.14 \times 10^{-4}$ | 46.21 |

As forest age increased, the coefficient of variation for each trait gradually decreased and tended to stabilize. As shown in Figure 1, the mean coefficient of variation for DBH (17.40%) was slightly lower than that for HT (23.9%) and much smaller than that for V (55.87%). These results indicated high potential for genetic improvement in V among the families in the *N. cadamba* progeny test population.

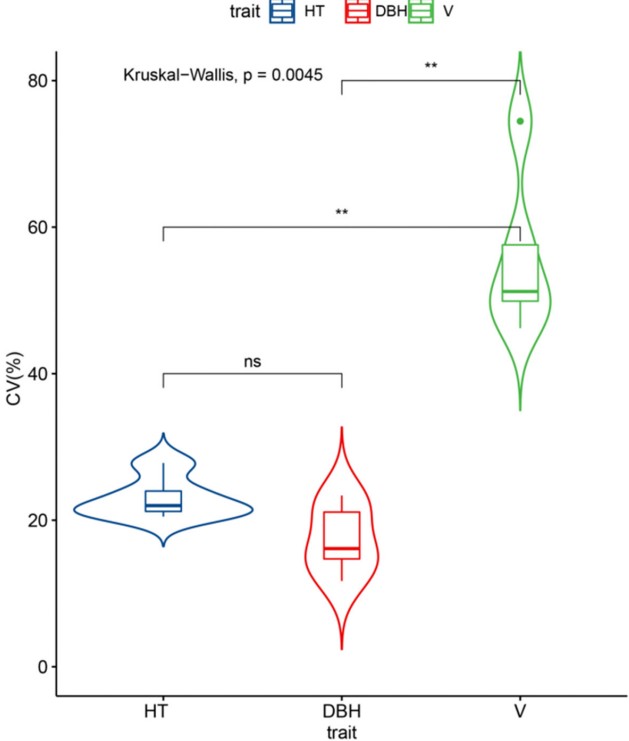

**Figure 1.** Coefficients of variation for traits. "**" indicates a significant difference at the 0.01 level, "ns" means not significant.

The results of Duncan's multiple range tests ($\alpha = 0.05$) among provenances of different forest ages showed that the differences of between provenances for each growth trait were significant (Table A1). At age 2 and age 3, the best performance of HT, DBH, and V was registered by provenance 10, while in the sixth year, the best performance of HT, DBH, and V was registered by provenance 3, indicating that the efficiency of early selection of *N. cadamba* is unstable and the time of early selection should not be too early.

### 3.2. Variance Analysis and Heritability Estimates

Using a linear mixed model with block as a fixed effect, and provenance, family within provenance, provenance–block interaction, and the family–block interaction as random effects, we subjected HT, DBH, and V at different ages to variance analysis (Table 3). ANOVA showed that the effects of block as a fixed effect on each trait were extremely significant ($p < 0.001$). For the random effects, the effects of provenance on each trait were significant, except the HT2 and V2, and family effects were significant in all traits except HT2. In general, the highly significant effects ($p < 0.001$) of provenance–block interaction were mainly observed at age 2, age 3, and age 4, and the significant effect of family–block interaction were mainly observed at age 2, age 3, age 4, and age 5. Except for the HT, with increasing of tree's age, the effect of interactive factors on the traits became smaller and smaller.

Additionally, the variance components and the heritability of the growth traits variances with age were estimated (Table 4). The provenance variance components ($V_P$) had no obvious age change trend. The family variance components ($V_F$) for DBH and V increased with increasing age, being similar to the variance components for individual tree within plot ($Ve$). The family variance components ($V_F$) for HT increased up to age 4 and substantially decreased at age 5. The variance component of provenance–block interaction ($V_{PB}$) for 60% traits was zero. With the increasing of forest age, the family–block interaction variance components ($V_{FB}$) for DBH and V first increased and then decreased, while there was no obvious change trend for HT.

Furthermore, the individual heritability was estimated from the second to the sixth year after planting for all trees measured in the field. The heritability estimates for DBH and V had a similar trend—both of them reached the lowest value in the second year and the highest in the fifth year. However, the minimum and maximum heritability of HT appeared in the sixth and fourth years, respectively. The mean estimated heritabilities for HT, DBH, and V across ages were 0.13, 0.19, and 0.18, respectively, indicating a moderate to low degree of additive genetic control.

The age trends for percentage family–block interaction variance relative to family variance ($V_{FB}/V_F$) showed a completely opposite pattern to that for the family variance components. The mean values of $V_{FB}/V_F$ for HT, DBH, and V were 8.69, 2.30, and 3.02, respectively. The mean value of $V_{FB}/V_F$ for HT was almost three times that of DBH and V, indicating that HT is more affected by the environment. The largest values of $V_{FB}/V_F$ for HT, DBH, and V were 17.91, 6.49, and 8.15, respectively, but they appeared at different ages. The largest values of $V_{FB}/V_F$ for DBH and V appeared in the second year, while the largest value of $V_{FB}/V_F$ for HT appeared in the sixth year (Table 4). This suggests that the effect of genotype by environment interaction varied greatly across traits.

The age trends in the genetic variation coefficients ($CV_G$) showed similar patterns for all three growth traits, first increasing and then decreasing. The means of the $CV_G$ for HT, DBH, and V across ages were 3.20%, 7.19%, and 12.98%, respectively. Throughout the measurement process, the genetic variation coefficient for V was always greater than those for both HT and DBH, indicating that V has a greater selection potential than HT and DBH.

**Table 3.** Variance analysis of each trait in a linear mixed effect model.

| Age | | | | Age 2 | | | Age 3 | | | Age 4 | | | Age 5 | | | Age 6 | | |
|---|---|---|---|---|---|---|---|---|---|---|---|---|---|---|---|---|---|---|
| Trait | Source | DF | | MeanSq | F Value | Pr(>F) | MeanSq | F Value | Pr(>F) | MeanSq | F Value | Pr(>F) | MeanSq | F Value | Pr(>F) | MeanSq | F Value | Pr(>F) |
| HT | Blk | 9 | | 3.16 | 11.67 | <0.0001 | 36.70 | 45.34 | <0.0001 | 38.31 | 40.43 | <0.0001 | 23.02 | 12.23 | <0.0001 | 11.95 | 9.17 | <0.0001 |
| | Pro | 10 | | 1.42 | 1.66 | 0.0859 | 9.51 | 4.90 | <0.0001 | 9.79 | 5.31 | <0.0001 | 7.11 | 2.43 | 0.0072 | 11.30 | 4.45 | <0.0001 |
| | Fam (Pro) | 28 | | 1.16 | 1.66 | 0.0172 | 3.04 | 1.56 | 0.0313 | 5.15 | 2.07 | 0.0009 | 4.45 | 1.50 | 0.0456 | 3.32 | 1.09 | 0.3412 |
| | Pro-blk | 90 | | 0.85 | 3.16 | <0.0001 | 1.94 | 2.40 | <0.0001 | 1.84 | 1.94 | <0.0001 | 2.92 | 1.55 | 0.0011 | 2.54 | 1.95 | <0.0001 |
| | Fam (Pro)-blk | 250 | | 0.70 | 2.57 | <0.0001 | 1.94 | 2.40 | <0.0001 | 2.49 | 2.63 | <0.0001 | 2.96 | 1.57 | <0.0001 | 3.04 | 2.33 | <0.0001 |
| | Error (within plot) | 1140 | | 0.27 | | | 0.81 | | | 0.95 | | | 1.88 | | | 1.30 | | |
| DBH | Blk | 9 | | 17.50 | 23.08 | <0.0001 | 51.67 | 20.76 | <0.0001 | 60.45 | 16.08 | <0.0001 | 89.25 | 20.63 | <0.0001 | 38.63 | 6.84 | <0.0001 |
| | Pro | 10 | | 5.14 | 2.44 | 0.0069 | 23.90 | 4.71 | <0.0001 | 31.36 | 6.49 | <0.0001 | 25.44 | 6.10 | <0.0001 | 23.76 | 3.29 | 0.0003 |
| | Fam (Pro) | 28 | | 3.04 | 1.80 | 0.0064 | 10.24 | 2.28 | <0.0001 | 12.09 | 2.55 | <0.0001 | 15.98 | 3.07 | <0.0001 | 16.76 | 2.95 | <0.0001 |
| | Pro-blk | 90 | | 2.10 | 2.78 | <0.0001 | 5.07 | 2.04 | <0.0001 | 4.83 | 1.28 | 0.0421 | 4.17 | 0.96 | 0.5766 | 7.22 | 1.28 | 0.0467 |
| | Fam (Pro)-blk | 250 | | 1.68 | 2.22 | <0.0001 | 4.48 | 1.80 | <0.0001 | 4.74 | 1.26 | 0.0074 | 5.21 | 1.20 | 0.0263 | 5.68 | 1.01 | 0.4675 |
| | Error (within plot) | 1140 | | 0.76 | | | 2.49 | | | 3.76 | | | 4.33 | | | 5.65 | | |
| V | Blk | 9 | | $2.11 \times 10^{-5}$ | 18.02 | <0.0001 | $1.26 \times 10^{-3}$ | 30.11 | <0.0001 | $4.11 \times 10^{-3}$ | 22.81 | <0.0001 | $9.10 \times 10^{-3}$ | 18.40 | <0.0001 | $6.42 \times 10^{-3}$ | 7.11 | <0.0001 |
| | Pro | 10 | | $6.68 \times 10^{-6}$ | 1.76 | 0.0630 | $4.38 \times 10^{-4}$ | 4.31 | <0.0001 | $1.40 \times 10^{-3}$ | 5.15 | <0.0001 | $2.45 \times 10^{-3}$ | 4.96 | <0.0001 | $4.16 \times 10^{-3}$ | 4.01 | <0.0001 |
| | Fam (Pro) | 28 | | $4.54 \times 10^{-6}$ | 1.64 | 0.0196 | $1.82 \times 10^{-4}$ | 2.31 | 0.0001 | $5.88 \times 10^{-4}$ | 2.26 | 0.0002 | $1.72 \times 10^{-3}$ | 2.84 | <0.0001 | $2.56 \times 10^{-3}$ | 2.72 | <0.0001 |
| | Pro-blk | 90 | | $3.79 \times 10^{-6}$ | 3.24 | <0.0001 | $1.02 \times 10^{-4}$ | 2.43 | <0.0001 | $2.71 \times 10^{-4}$ | 1.50 | 0.0023 | $4.95 \times 10^{-4}$ | 1.00 | 0.4814 | $1.04 \times 10^{-3}$ | 1.15 | 0.1696 |
| | Fam (Pro)-blk | 250 | | $2.77 \times 10^{-6}$ | 2.37 | <0.0001 | $7.87 \times 10^{-5}$ | 1.88 | <0.0001 | $2.60 \times 10^{-4}$ | 1.44 | <0.0001 | $6.06 \times 10^{-4}$ | 1.23 | 0.0169 | $9.42 \times 10^{-4}$ | 1.04 | 0.3270 |
| | Error (within plot) | 1140 | | $1.17 \times 10^{-6}$ | | | $4.19 \times 10^{-5}$ | | | $1.80 \times 10^{-4}$ | | | $4.94 \times 10^{-4}$ | | | $9.02 \times 10^{-4}$ | | |

Notes: HT, DBH, V, Blk, Pro, Fam (Pro), Pro-blk, and Fam (Pro)-blk represent tree height, diameter at breast height, tree volume, block, provenance, family within provenance, provenance–block interaction, and family–block interaction, respectively.

**Table 4.** Variance components for growth traits at different ages.

| Trait | $V_P$ | $V_F$ | $V_{PB}$ | $V_{FB}$ | Ve | $V_{FB}/V_F$ | $h^2$ | $CV_G$(%) |
|---|---|---|---|---|---|---|---|---|
| HT2 | 0.0030 | 0.0128 | 0.0101 | 0.1115 | 0.2739 | 8.71 | 0.13 | 3.26 |
| HT3 | 0.0705 | 0.0320 | 0.0000 | 0.2663 | 0.8157 | 8.33 | 0.11 | 4.14 |
| HT4 | 0.0261 | 0.0887 | 0.0000 | 0.3253 | 0.9552 | 3.67 | 0.26 | 3.48 |
| HT5 | 0.0158 | 0.0474 | 0.0028 | 0.2299 | 1.9250 | 4.85 | 0.09 | 2.31 |
| HT6 | 0.0987 | 0.0230 | 0.0000 | 0.4115 | 1.3225 | 17.91 | 0.05 | 2.86 |
| DBH2 | 0.0142 | 0.0380 | 0.0221 | 0.2467 | 0.7645 | 6.49 | 0.14 | 8.31 |
| DBH3 | 0.1165 | 0.1410 | 0.0000 | 0.4938 | 2.5074 | 3.50 | 0.18 | 9.31 |
| DBH4 | 0.1568 | 0.1829 | 0.0000 | 0.2003 | 3.7957 | 1.10 | 0.18 | 7.49 |
| DBH5 | 0.0667 | 0.2948 | 0.0000 | 0.1290 | 4.3506 | 0.44 | 0.25 | 5.80 |
| DBH6 | 0.0451 | 0.2986 | 0.0745 | 0.0000 | 5.6961 | 0.00 | 0.20 | 5.04 |
| V2 | 0.00 | $5.35 \times 10^{-8}$ | $5.38 \times 10^{-8}$ | $4.36 \times 10^{-7}$ | $1.18 \times 10^{-6}$ | 8.15 | 0.13 | 12.93 |
| V3 | $2.23 \times 10^{-6}$ | $2.49 \times 10^{-6}$ | 0.00 | $1.01 \times 10^{-5}$ | $4.19 \times 10^{-5}$ | 4.06 | 0.18 | 15.85 |
| V4 | $6.86 \times 10^{-6}$ | $8.36 \times 10^{-6}$ | 0.00 | $1.90 \times 10^{-5}$ | $1.81 \times 10^{-4}$ | 2.27 | 0.16 | 13.09 |
| V5 | $4.00 \times 10^{-6}$ | $3.09 \times 10^{-5}$ | 0.00 | $1.96 \times 10^{-5}$ | $4.96 \times 10^{-4}$ | 0.63 | 0.23 | 12.09 |
| V6 | $8.28 \times 10^{-6}$ | $4.77 \times 10^{-5}$ | $7.18 \times 10^{-6}$ | 0.00 | $9.12 \times 10^{-4}$ | 0.00 | 0.20 | 10.96 |

Notes: $V_P$, $V_F$, $V_{PB}$, $V_{FB}$, $V_e$, $h^2$, and $CV_G$ represent the variance components of provenance, family, provenance–block interaction, family–block interaction, residual error, individual heritability, and the genetic variation coefficient, respectively.

The components of variance due to different sources were estimated as percentages of the total phenotypic variance for each studied growth trait (Figure 2). The largest proportion of variance component was always the $Ve$, which accounted for 66.61% to 93.52% of the total variance. However, provenance variance components accounted for 0 to 5.95%, and the family variance components 1.24% to 6.35%. Interestingly, the proportion of family variance components was always larger than the provenance variance components, except the HT3 and HT6. The average percentage of provenance variance components for HT, DBH, and V were 2.92%, 2.12%, and 1.74%, respectively, while the average percentage of family variance components for HT, DBH, and V were 3.11%, 4.60%, and 4.38%, respectively, which indicated that the variation among families within provenance is greater than that among provenance in *N. cadamba*. Similarly, the proportion of variance due to the family–block interaction (0–27.11%) was always larger than the provenance–block variance components (0–3.13%), except for the DBH6 and V6. It should be noted that the changes over time in the family–block interaction were not consistent across the three traits. For DBH and V, they gradually decreased from the second year (15.15% and 25.37%, respectively) and decreased to 0 in the sixth year. However, there was no obvious trend in proportion of family variance components of HT, although they were always larger than 10%. This suggests that the effects of the genotype × environment interaction for DBH and V decreased with age, but that for HT exhibited no clear trend.

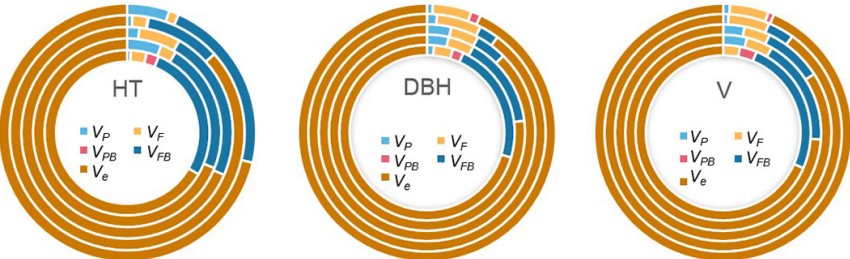

**Figure 2.** Variance components of different factors as percentages of phenotypic variance for growth traits at different ages in a progeny test population of *N. cadamba*. From the inside to the outside, the order is from age 2 to age 6. $V_P$, $V_F$, $V_{PB}$, $V_{FB}$, and $V_e$ represent the variance components for provenance, family, provenance–block interaction, family–block interaction, and residual error, respectively.

### 3.3. Age–Age Genetic and Phenotypic Correlations

The additive genetic correlations between the early age and the reference age (age 6) for the three growth traits were always strong ($r_A > 0.50$, Table 5). DBH and V had stronger additive genetic correlations than HT, but this relationship was unstable when comparing the age–age additive genetic correlations for HT. Because the pairs of measurements became much closer over time, the additive genetic correlations for DBH and V both increased as the early age increased. However, no such trend was observed for HT. This may have been due to the relatively strong effect of the genotype × environment interaction for HT. The standard errors of the additive genetic correlation estimates for all three traits were between 0.01 and 0.21. The standard error was always highest for the additive genetic correlations between the ages of 2 and 6, possibly because of the use of different sampling and measurement methods, as well as the larger environmental effect on early measurements due to planting shock.

**Table 5.** Genetic and phenotypic correlations between the early age and the reference age (age 6) for the three growth traits.

| Trait | Early Age | $r_A$ | $r_P$ | $r_A$-$r_P$ |
|:-----:|:---------:|:-----:|:-----:|:-----------:|
| HT | 2 | $0.51 \pm 0.21$ | $0.12 \pm 0.02$ | +0.39 |
| | 3 | $0.87 \pm 0.08$ | $0.47 \pm 0.02$ | +0.40 |
| | 4 | $0.77 \pm 0.12$ | $0.58 \pm 0.07$ | +0.19 |
| | 5 | $0.73 \pm 0.18$ | $0.58 \pm 0.09$ | +0.15 |
| DBH | 2 | $0.79 \pm 0.16$ | $0.10 \pm 0.03$ | +0.69 |
| | 3 | $0.89 \pm 0.06$ | $0.65 \pm 0.02$ | +0.24 |
| | 4 | $0.95 \pm 0.03$ | $0.81 \pm 0.01$ | +0.14 |
| | 5 | $0.99 \pm 0.01$ | $0.90 \pm 0.01$ | +0.09 |
| V | 2 | $0.74 \pm 0.17$ | $0.12 \pm 0.03$ | +0.62 |
| | 3 | $0.92 \pm 0.05$ | $0.64 \pm 0.02$ | +0.28 |
| | 4 | $0.96 \pm 0.03$ | $0.82 \pm 0.01$ | +0.14 |
| | 5 | $0.98 \pm 0.01$ | $0.92 \pm 0.01$ | +0.06 |

The age-age phenotype correlations for all growth traits increased with time, but the estimated phenotypic correlations were generally smaller than the corresponding estimated genetic correlations ($r_A$-$r_P$ > 0). In addition, the age–age phenotypic correlations for DBH and V were markedly greater than those for HT, except age 2.

The relationships of *LAR* to the age–age genetic correlations for the analyzed traits are shown in Figure 3. The estimated magnitudes of the regression slope for DBH and V (0.228 and 0.264, respectively) were larger than that for HT (0.221), and the $R^2$ values of the linear regressions for DBH and V (0.9989 and 0.8915, respectively) were greater than that for HT (0.3465). This suggests a possibility of early selection for DBH and V. In accordance with expectations and our previous findings, the age–age genetic correlations identified using the linear model increased as the time interval between the early and reference ages decreased.

As Figure 4 shows, the estimated regression slopes for the phenotypic correlations were 0.868 for DBH, 0.513 for HT, and 0.876 for V, and were thus greater than those for the genetic correlations (Figure 3). These phenotypic correlations were described well by linear models with *LAR* as the independent variable; the corresponding $R^2$ values were 0.9317 for DBH, 0.8694 for HT, and 0.9453 for V. For DBH, the $R^2$ value for the linear model relating the additive genetic correlation ($r_A$) to the *LAR* (0.9989) was higher than that for phenotypic correlation (0.9317). In contrast, the $R^2$ values for the linear models relating the phenotypic correlations for HT and V to *LAR* (0.8694 and 0.9453, respectively) were larger than those for the corresponding additive genetic correlations (0.3465 and 0.8915). This indicated that when using linear models based on *LAR* to predict age–age correlations in growth traits for ages outside the studied range, the best results were achieved for DBH when focusing on genetic correlations, while the best results for HT and V were achieved when focusing on phenotypic correlations.

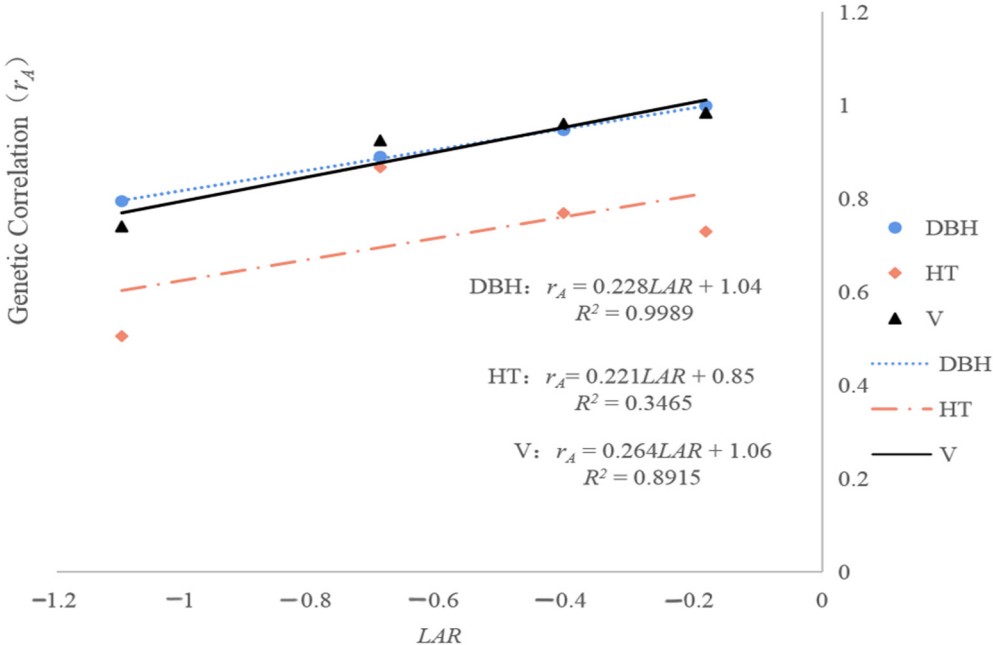

**Figure 3.** Relationship of *LAR* ($log_e(\text{age}_{\text{young}}/\text{age}_{\text{old}})$) to age–age genetic correlations.

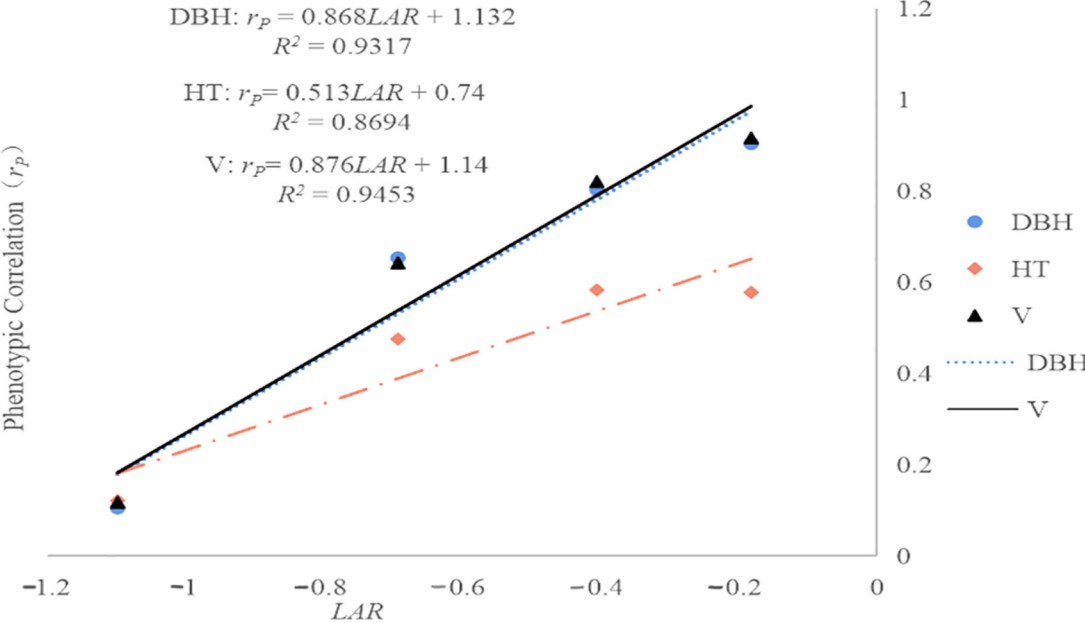

**Figure 4.** Relationship of *LAR* ($log_e(\text{age}_{\text{young}}/\text{age}_{\text{old}})$) to age–age phenotypic correlations.

With increasing forest age, the extent to which the phenotypic correlation and genetic correlation coefficients increased differently; at age 2, the genetic correlation coefficients for each growth trait were 0.54–0.79, while the corresponding phenotypic correlation coefficients were only 0.10–0.12. This is consistent with the hypothesis that the efficiency of selection increases with forest age.

### 3.4. Efficiencies of Early Selection

Figure 5 shows the efficiencies achieved through early selection ($SE_{GPY}$) for HT, DBH, and V at a rotation age of 12. For all three growth traits (with the exception of V at age 4), selection efficiency increased strongly with age. Selection efficiency was highest at age 5 (157.28%, 151.56%, and 127.08% for V, DBH, and HT). The efficiency for DBH was much

higher than that for HT and V in the first three years, while V showed the highest efficiency at age 5. This suggests that the greatest annual gains are achieved by performing direct selection of growth trait on relatively old trees, and the growth potential of *N. cadamba* is not fully expressed during the period of early growth. Consequently, performing selection on *N. cadamba* less than 4 years of age may have serious disadvantages.

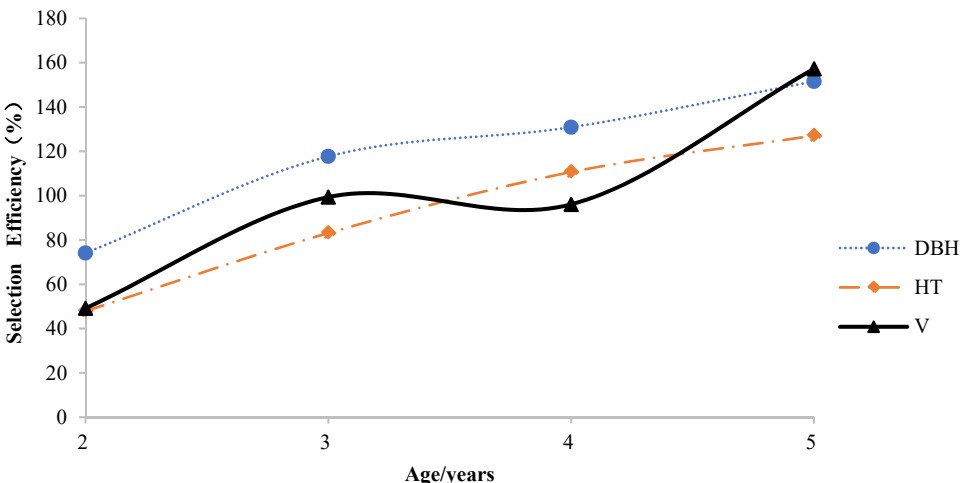

**Figure 5.** Selection efficiency for the three growth traits at various selection ages, taking growth traits at age 12 as the targets for improvement.

### 3.5. Early Selection for Provenances and Families

According to the results of early selection efficiency, we selected the data of the fifth year as the best age for early selection, V as the selection index, selection intensity positioning 30%, 3 superior provenances, and 12 superior families (Table 6). Taking three times of the standard deviation as the selection intensity of individual selection in each trait of the superior family, we selected 129 plus trees from 12 superior families, with a selection rate of 21.50%.

**Table 6.** Genetic gain of selection at different levels of variation.

| Selection Level | Parameter | HT (m) | DBH (cm) | V (m³) |
|---|---|---|---|---|
| Provenance | Selected mean | 10.67 | 11.84 | $5.72 \times 10^{-2}$ |
| | Total mean | 10.34 | 11.04 | $5.03 \times 10^{-2}$ |
| | Realized gain (%) | 3.14 | 7.18 | 13.66 |
| Family | Selected mean | 10.67 | 11.55 | $5.65 \times 10^{-2}$ |
| | Total mean | 10.35 | 10.85 | $4.88 \times 10^{-2}$ |
| | Realized gain (%) | 3.08 | 6.39 | 15.67 |
| Individual | Selected individual mean | 10.67 | 11.55 | $5.65 \times 10^{-2}$ |
| | Selected families mean | 14.18 | 11.93 | $8.91 \times 10^{-2}$ |
| | Realized gain (%) | 11.77 | 22.77 | 57.61 |
| | Genetic gain (%) | 0.60 | 2.43 | 4.77 |

Regardless of whether they were from provenance, family, or individual levels, the realized gain of V was the largest and that of HT was the smallest. The genetic gain of individual level selection had the same pattern. The realized gain of DBH and HT for the provenance level was slightly higher than that for the family level, while the realized gain of V for the provenance level was slightly lower than that for family level. It is worth noting that the realized gain of all traits at the individual level was much higher than that at the provenance level and the family level.

## 4. Discussion

Tree height, diameter at breast height, and tree volume are important traits of forest tree growth that determine the economic value of forest trees and are therefore of great interest to forest tree breeders [6]. The results presented here demonstrated significant genetic variation in HT, DBH, and V among half-sib families and showed that family selection in the progeny test population of *N. cadamba* could be used to genetically improve these traits. It was previously reported that *N. cadamba* trees up to 5 years old in South Kalimantan exhibited DBH growth of 1.2–4.8 cm/year and height growth of 0.8–3.7 m/year, and the growth rates for both variables were higher in Java than that in South Kalimantan [28]. The mean annual increment (MAI) in HT, DBH, and V for *N. cadamba* trees up to 6 years old in the trial site in the present study were 1.12–3.88 m/year, 1.28–2.70 cm/year, and 0.0119–0.0194 m$^3$/year, respectively. The rates of increase in HT and DBH were thus similar to those in South Kalimantan. The differences in growth rates between the two studies were probably due to differences in genotypes and site quality. From age 3 to age 6, the mean annual increment for HT and DBH decreased while that for V increased (Table 2). It should be noted that the mean annual increment in HT declined more rapidly with age than that for DBH. These trends are consistent with the findings of Krisnawati et al. [28]. However, in other tree species, including *Larix principis-rupprechtii* [30], shore pine [31], and lodgepole pine [32], these declines in the rates of mean annual increment for HT and DBH followed different trends, probably due to differences among tree species.

Site productivity is maximized when the rotation age coincides with the peak of the mean annual increment [33]. Table 2 shows that the mean annual increment of V did not reach a peak in the age range examined in this study. Together with the above variance analysis, it is suggested that the ideal rotation age for *N. cadamba* is greater than six years. These findings are consistent with those of Wei and Zhu [19].

The variance components of growth traits have generally been found to increase with age in studies on coniferous trees including radiata pine [34], Scots pine [35], Douglas fir [36], and *Larix principis-rupprechtii* [30]. The family variance components of DBH and V displayed similar trends in this work, while the provenance variance components had no obvious trend with age (Table 4, Figure 2). Although the family variance components for HT first increased with age but then decreased, this may have been due to family–block interactions. For all three traits, the percentage of family variance components were always greater than the percentage of provenance variance components, indicating that the variation among families within provenance is greater than that among provenance in *N. cadamba*. A same pattern was found in another study in *N. cadamba* [37], and a similar pattern was found in eucalyptus [5]. Additionally, the proportion of variance due to the family–block interaction was also larger than the provenance–block variance components. It should be noted that the mean value of $V_{FB}/V_F$ for HT was almost three times that of DBH and V, indicating that HT is more affected by the environment. This may have been due to the fact that the growth of HT is more affected by leaf area after the forest was closed. Large effects of genotype $\times$ environment interactions on growth traits have been reported in other studies [29,38,39].

Heritability reflects the degree of genetic control exerting on growth traits. The estimated individual heritability values for the growth traits investigated here ranged from 0.05 to 0.26, indicating that the variation in *N. cadamba* growth traits was subject to weak or intermediate genetic control. These heritability values are lower than those obtained in an earlier study [40]. The reason for this deviation may be that different forms of heritability were used in the two studies; the earlier study computed broad sense heritability, whereas the additive heritability was examined in this study. The heritabilities of different growth traits showed different trends with increasing age, in accordance with previous studies on tree species including sugi (*Cryptomeria japonica*) [7], *Larix kaempferi* [29], and *Larix principis-rupprechtii* [30], showing that the individual heritability of DBH changed less with age than that of HT, which may be due to the fact that DBH receives less environmental impact than HT.

$CV_G$ measures genetic variance after standardization against the trait mean and is considered the best parameter to use when comparing genetic variation [41]. The mean $CV_G$ for V was found to be almost twice that for DBH and four times that for HT, in keeping with earlier studies on *Larix kaempferi* [42] and jack pine (*Pinus banksiana*) [43]. At any given age, the $CV_G$ for V was always the largest, and the $CV_G$ for HT was always the smallest, suggesting that the selection potential for V is greater than that for DBH, which in turn exceeds that for HT. The $CV_G$ for growth traits decreased with increasing age; a similar trend with respect to the additive coefficient of variation ($CV_A$) has been reported previously [27,35].

Age–age genetic correlations are widely used to determine the optimal age of selection in genetic improvement programs [44]. We found that the genetic correlations between early and late ages for growth traits were always positive and high (0.51–0.99), demonstrating the potential for *N. cadamba* selection at a young age. Age–age genetic correlations for both DBH and V were higher than that for HT, in agreement with observations on *Larix kaempferi* [42] and sugi [7] but not with those of Dong [30], who found that the genetic correlation between early and late ages for HT was higher than that for DBH. It is not clear why the relative magnitudes of the genetic correlations for growth traits differed between these studies. Potential contributing factors include differences in the studied species, competitive pressures, growth phases, and silvicultural factors, but further investigation is needed to clarify this issue [30]. The most widely used model for exploring age–age genetic correlations is that of Lambeth (1980), which has been tested by several researchers working on tree species [7,30,39,45]. The age–age genetic correlations were generally stronger than the corresponding phenotypic correlations, in agreement with observations on *Larix principis-rupprechtii* [30]. Predictive models using the age–age genetic correlations achieved excellent fits for both DBH and V ($R^2 > 0.86$) for the models using *LAR*, while it did not fit well for HT (0.35). In addition, our analysis of temporal trends in genetic parameters for growth traits revealed that DBH and V were more advantageous than HT in determining early selection age. Thus, DBH and V may be more effective criteria than HT for the early selection of *N. cadamba* if genetic gains are to be maximized.

Selection efficiency is a statistic that combines information on both genetic parameters and time discount factors [46]. It is generally considered that the optimal age for early selection is the age at which the efficiency of early selection peaks [47]. Selection efficiencies were found to initially increase with age and then decrease in *Larix principis-rupprechtii* [30] and *Pinus taeda* [39], while efficiency decreased with age but then increased in *Larix kaempferi* [42]. In the present study, selection efficiency increased markedly with age. The differences between this finding and previous reports may be due to species differences. Our results suggest that the optimum age for early selection before half-rotation in *N. cadamba* is age 5. The findings presented here will provide useful information to guide early selection efforts in *N. cadamba* improvement programs in southern China.

Through the comparison of the realized gain obtained by different selection levels, the breeding strategy of selecting plus trees from superior families can obtain a greater realized gain. This strategy of combined selection is often used in other tree species [4,39,48]. These plus trees selected in the present study can be used as the parent of the next generation. In addition, in this study, it was also found that the provenance introduced from abroad (provenance 13) grew well in Southern China and was selected as an excellent provenance, suggesting that more foreign provenances should be introduced to supplement germplasm resources in the follow-up breeding work in *N. cadamba*.

The age-dependence of genetic parameters for growth traits in *N. cadamba* has not previously been investigated to our knowledge. Genetic parameters such as variance components and heritability reflect the degree to which genetic control account for trait variation. By studying age trends in genetic parameters, one can characterize the growth patterns of trees such as *N. cadamba*, and the results obtained can provide guidance for subsequent breeding and cultivation efforts.

## 5. Conclusions

The multi-level genetic variation and selection efficiency for growth traits in *N. cadamba* were analyzed in this study. The obtained results indicated that the family effect had higher contribution than provenance effect to the growth traits. The study also revealed the feasibility of early selection for growth in *N. cadamba* is best performed at age 5. In general, these results can be expected to provide theoretical guidance and materials for breeding programs in *N. cadamba* and accelerate the breeding process. In addition, this study can provide a reference for the breeding strategies of other fast-growing tree species.

**Author Contributions:** Conceptualization, X.C. and K.O.; methodology, Q.Q.; software, Q.Q.; validation, C.L., B.L. and H.S.; formal analysis, K.O. and R.P.; investigation, Q.Q., C.L., B.L., H.S. and H.L.; resources, X.C., K.O. and H.L.; data curation, Q.Q. and C.L.; writing—original draft preparation, Q.Q.; writing—review and editing, P.L., R.P. and K.O.; visualization, Q.Q.; supervision, X.C. and K.O.; project administration, P.L., R.P. and K.O.; funding acquisition, X.C. and K.O. All authors have read and agreed to the published version of the manuscript.

**Funding:** This research was funded by the Science and Technology Program of Guangdong, China (grant number 2017B020201008); Forestry Science and Technology Innovation Project in Guangdong Province (grant number 2019KJCX001); National Natural Science Foundation of China (grant number 31600525); Extension and Demonstration Project of Forest Science and Technology from China State Financial Budget ((2018)GDTK-08); Science and Technology Program of Guangzhou (grant number 201904020014); 2018 Big Pig-producing County Reward Funds (research and promotion of key technologies for healthy feeding of pigs and resource utilization of manure pollution).

**Institutional Review Board Statement:** Not applicable.

**Informed Consent Statement:** Not applicable.

**Data Availability Statement:** The data presented in this study are available on request from the correspondence authors.

**Acknowledgments:** We thank Yue Li for his help in valuable discussion in our study.

**Conflicts of Interest:** The authors declare no conflict of interest.

# Appendix A

**Table A1.** Statistics of growth traits for all provenances at different forest ages.

| Trait | Provenance | Age 2 | Age 3 | Age 4 | Age 5 | Age 6 |
|---|---|---|---|---|---|---|
| HT | 1 | 3.81 ± 0.51 c | 7.86 ± 0.85 abcd | 9.96 ± 1.19 bcd | 11.02 ± 1.68 bcd | 11.92 ± 1.22 cde |
| | 3 | 3.88 ± 0.70 bc | 8.07 ± 1.18 abc | 10.84 ± 0.99 a | 11.56 ± 1.40 ab | 13.12 ± 0.92 a |
| | 4 | 3.87 ± 0.62 bc | 7.85 ± 1.24 abcd | 9.86 ± 1.14 bcde | 11.30 ± 1.59 abc | 12.57 ± 1.41 abcde |
| | 5 | 3.57 ± 0.58 c | 7.51 ± 1.15 cd | 9.40 ± 1.26 def | 10.28 ± 1.53 d | 11.67 ± 1.38 e |
| | 6 | 3.80 ± 0.42 c | 6.63 ± 1.03 e | 9.04 ± 1.21 f | 10.50 ± 1.52 cd | 11.83 ± 1.45 cde |
| | 7 | 3.68 ± 0.67 c | 7.24 ± 1.20 d | 9.20 ± 1.38 ef | 10.39 ± 1.59 d | 11.73 ± 1.42 de |
| | 8 | 3.90 ± 0.66 bc | 7.78 ± 1.08 abcd | 9.91 ± 1.17 bcde | 10.85 ± 1.44 bcd | 12.13 ± 1.29 bcde |
| | 9 | 3.86 ± 0.65 bc | 7.68 ± 1.20 bcd | 9.55 ± 1.30 cdef | 10.77 ± 1.53 bcd | 12.16 ± 1.38 abcde |
| | 10 | 4.41 ± 0.64 a | 8.43 ± 1.03 a | 10.26 ± 1.18 abc | 11.40 ± 1.33 ab | 12.71 ± 1.23 abcd |
| | 11 | 3.93 ± 0.61 bc | 8.32 ± 0.98 ab | 10.34 ± 1.08 ab | 11.29 ± 1.55 abc | 12.78 ± 1.23 abc |
| | 13 | 4.24 ± 0.64 ab | 8.28 ± 1.10 ab | 10.71 ± 1.00 a | 12.11 ± 1.42 a | 13.06 ± 1.41 ab |
| DBH | 1 | 2.58 ± 0.92 bc | 5.21 ± 1.65 d | 7.44 ± 2.02 c | 9.94 ± 2.39 de | 11.08 ± 2.62 ef |
| | 3 | 2.69 ± 1.17 bc | 5.68 ± 2.04 ab | 8.03 ± 1.76 ab | 10.84 ± 2.27 a | 12.59 ± 2.22 a |
| | 4 | 2.61 ± 1.18 bc | 5.26 ± 2.05 cd | 7.68 ± 2.36 bc | 10.38 ± 2.23 abcd | 11.59 ± 2.32 cd |
| | 5 | 2.53 ± 0.97 c | 5.22 ± 1.77 d | 7.41 ± 1.94 c | 10.10 ± 2.27 cde | 11.46 ± 2.47 def |
| | 6 | 2.61 ± 1.01 bc | 4.57 ± 1.69 e | 6.83 ± 2.35 d | 9.68 ± 2.18 e | 11.00 ± 2.71 f |
| | 7 | 2.78 ± 1.08 abc | 5.26 ± 1.82 cd | 7.70 ± 2.14 bc | 10.25 ± 2.28 bcd | 11.54 ± 2.55 cde |
| | 8 | 2.82 ± 1.04 ab | 5.60 ± 1.66 abcd | 7.94 ± 1.93 ab | 10.41 ± 2.2 abcd | 11.72 ± 2.45 bcd |
| | 9 | 2.78 ± 1.08 abc | 5.40 ± 1.96 bcd | 7.74 ± 2.26 bc | 10.30 ± 2.38 abcd | 11.49 ± 2.54 cdef |
| | 10 | 2.94 ± 1.10 a | 5.86 ± 1.70 a | 8.23 ± 2.03 a | 10.62 ± 2.34 abc | 11.97 ± 2.53 bcd |
| | 11 | 2.73 ± 1.02 abc | 5.78 ± 1.67 ab | 8.02 ± 1.91 ab | 10.70 ± 2.1 ab | 11.98 ± 2.23 bc |
| | 13 | 2.81 ± 1.17 ab | 5.63 ± 2.01 abc | 7.93 ± 2.23 ab | 10.54 ± 2.29 abc | 12.13 ± 2.78 ab |
| V | 1 | $1.56 \times 10^{-3} \pm 1.02 \times 10^{-3}$ c | $1.30 \times 10^{-2} \pm 6.62 \times 10^{-3}$ cd | $2.95 \times 10^{-2} \pm 1.44 \times 10^{-2}$ cd | $4.87 \times 10^{-2} \pm 2.50 \times 10^{-2}$ bcd | $6.24 \times 10^{-2} \pm 2.94 \times 10^{-2}$ c |
| | 3 | $1.83 \times 10^{-3} \pm 1.58 \times 10^{-3}$ bc | $1.56 \times 10^{-2} \pm 9.07 \times 10^{-3}$ abc | $3.66 \times 10^{-2} \pm 1.47 \times 10^{-2}$ a | $5.66 \times 10^{-2} \pm 2.32 \times 10^{-2}$ ab | $8.24 \times 10^{-2} \pm 2.81 \times 10^{-2}$ a |
| | 4 | $1.76 \times 10^{-3} \pm 1.32 \times 10^{-3}$ bc | $1.39 \times 10^{-2} \pm 8.57 \times 10^{-3}$ bcd | $3.05 \times 10^{-2} \pm 1.59 \times 10^{-2}$ cd | $5.26 \times 10^{-2} \pm 2.50 \times 10^{-2}$ abc | $7.18 \times 10^{-2} \pm 3.09 \times 10^{-2}$ abc |
| | 5 | $1.40 \times 10^{-3} \pm 1.04 \times 10^{-3}$ c | $1.24 \times 10^{-2} \pm 7.59 \times 10^{-3}$ d | $2.63 \times 10^{-2} \pm 1.31 \times 10^{-2}$ de | $4.33 \times 10^{-2} \pm 2.32 \times 10^{-2}$ d | $6.18 \times 10^{-2} \pm 2.98 \times 10^{-2}$ c |
| | 6 | $1.58 \times 10^{-3} \pm 9.61 \times 10^{-4}$ c | $8.51 \times 10^{-3} \pm 5.49 \times 10^{-3}$ e | $2.33 \times 10^{-2} \pm 1.44 \times 10^{-2}$ e | $4.20 \times 10^{-2} \pm 2.12 \times 10^{-2}$ d | $6.15 \times 10^{-2} \pm 3.21 \times 10^{-2}$ c |
| | 7 | $1.68 \times 10^{-3} \pm 1.39 \times 10^{-3}$ c | $1.18 \times 10^{-2} \pm 7.21 \times 10^{-3}$ d | $2.69 \times 10^{-2} \pm 1.53 \times 10^{-2}$ de | $4.47 \times 10^{-2} \pm 2.36 \times 10^{-2}$ cd | $6.34 \times 10^{-2} \pm 3.16 \times 10^{-2}$ c |
| | 8 | $1.85 \times 10^{-3} \pm 1.33 \times 10^{-3}$ bc | $1.38 \times 10^{-2} \pm 7.18 \times 10^{-3}$ bcd | $3.09 \times 10^{-2} \pm 1.42 \times 10^{-2}$ bcd | $4.86 \times 10^{-2} \pm 2.31 \times 10^{-2}$ bcd | $6.79 \times 10^{-2} \pm 3.12 \times 10^{-2}$ bc |
| | 9 | $1.82 \times 10^{-3} \pm 1.35 \times 10^{-3}$ bc | $1.36 \times 10^{-2} \pm 8.26 \times 10^{-3}$ bcd | $2.89 \times 10^{-2} \pm 1.5 \times 10^{-2}$ cd | $4.82 \times 10^{-2} \pm 2.55 \times 10^{-2}$ bcd | $6.73 \times 10^{-2} \pm 3.22 \times 10^{-2}$ bc |
| | 10 | $2.41 \times 10^{-3} \pm 1.46 \times 10^{-3}$ a | $1.67 \times 10^{-2} \pm 7.88 \times 10^{-3}$ a | $3.41 \times 10^{-2} \pm 1.58 \times 10^{-2}$ abc | $5.48 \times 10^{-2} \pm 2.58 \times 10^{-2}$ ab | $7.54 \times 10^{-2} \pm 3.23 \times 10^{-2}$ ab |
| | 11 | $1.80 \times 10^{-3} \pm 1.24 \times 10^{-3}$ bc | $1.60 \times 10^{-2} \pm 7.38 \times 10^{-3}$ ab | $3.36 \times 10^{-2} \pm 1.42 \times 10^{-2}$ abc | $5.36 \times 10^{-2} \pm 2.33 \times 10^{-2}$ abc | $7.56 \times 10^{-2} \pm 2.97 \times 10^{-2}$ ab |
| | 13 | $2.22 \times 10^{-3} \pm 1.72 \times 10^{-3}$ ab | $1.60 \times 10^{-2} \pm 9.27 \times 10^{-3}$ ab | $3.61 \times 10^{-2} \pm 1.67 \times 10^{-2}$ ab | $6.01 \times 10^{-2} \pm 2.52 \times 10^{-2}$ a | $8.06 \times 10^{-2} \pm 3.55 \times 10^{-2}$ a |

Note: Different lowercase letters after the same column data indicate significant difference ($p < 0.05$).

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
