# Peer review of "Multi-Level Genetic Variation and Selection Strategy of Neolamarckia cadamba in Successive Years"

_forests, doi:10.3390/f12111455_

Round 1

Reviewer 1 Report

  • The abstract must include data regarding the critical finds by the authors in terms of data of important findings. 
  • The introduction must have a clear hypothesis and significantly develop the second paragraph of this manuscript.
  • Overall there is the repetition of the information which could be avoided.  
  •  Check figure ligands; they are carelessly written.
  • Page 3 second last paragraph can be divided into understandable sentences. 
  • Discussion should include more information and references related to the relevant and related works. 
  • Restructure and carefully edit the conclusion section, especially in the conclusion section.

Reviewer 2 Report

Dear Authors,

I have read the manuscript “Multi-level genetic variation and selection improvement strategy of Neolamarckia cadamba in successive years” several times and, in my opinion, the paper can be interesting for the scientific community. The results are interesting. However, I have some recommendations, which in my opinion will increase the scientific soundness and help the reader to understand better the new information brought by the paper. I tried to make some improvements directly to the article (pdf). Unfortunately, I have to recommend a minor revision of the manuscript, especially because an extensive editing of the English language and style is required.

General Comments:

  1. The major weakness of the article is related to the necessity of an extensive editing of the English language. Also, it seems quite rudimentary to start each time the phrase with Figure 3 shows, Figure ... shows ... Table ... shows ... It is not advisable to present results (values) or references to tables in the discussion section and to compare with other species (we don't even know their age or whether the environmental conditions are similar). Reduce the Discussion section by briefly presenting the comparison with other species only if they have the same age and were growth in similar environmental conditions. In Bibliography, the authors must respect the Instructions for authors of Forests journal (E.g.: ‘;’ between authors, publication year with bold, volume with italic, the abbreviated name of journals, etc).

Specific comments:

Title

I suggest to delete ‘improvement’ from the title.

Abstract

- Line 24: Replace ‘The genetic correlations between ages’ with ‘The age-age genetic correlations‘.

- Line 29: I suggest to delete ‘breeding’ written before ‘materials’ to avoid repetition.

Introduction

- Line 42: ‘rotation period’ instead of  ‘period of rotation’.

- Lines 56-62: Recommendations directly in pdf.

- Lines 63-64: Rubiaceae with italic.

- Lines 72-90: Recommendations directly in pdf.

Materials and Methods

- Lines 93-108: Recommendations directly in pdf. The sentences underlined in yellow will be clarified after English editing.

- Lines 110-111: Minor changes in table 1.

- Line 129: where are VP and VPB in equation 3?

Results

- Lines 171-173: replace ‘and’ with ‘at’, add ‘years’ after ‘6’ and replace ‘by’ with ‘with’.

- Line 177: Insert a blank row after the table.

- Lines 186-347: Numerous recommendations, visible in pdf.

Discussion

- Lines 350-449: Numerous recommendations, visible in pdf. Reduce this section by giving up the comparison with other species or present this briefly if they have the same age and similar environmental conditions.

Conclusions

- Line 473: Please add ‘that’ after ‘indicating’.

- Line 479: Delete ‘breeding’ written before ‘materials’.

- Line 480: Delete ‘And’ and start the sentence with ‘In addition, this study can provide…’.

References

- Lines 505-611: You must respect the Instructions for authors of Forests journal (E.g.: ‘;’ between authors, publication year with bold, volume with italic, abbreviated name of journals, etc).
